# Human Melanocortin-2 Receptor: Identifying a Role for Residues in the TM4, EC2, and TM5 Domains in Activation and Trafficking as a Result of Co-Expression with the Accessory Protein, Mrap1 in Chinese Hamster Ovary Cells

**DOI:** 10.3390/biom12101422

**Published:** 2022-10-04

**Authors:** Perry V. Davis, Ciaran A. Shaughnessy, Robert M. Dores

**Affiliations:** Department of Biological Sciences, University of Denver, Denver, CO 80210, USA

**Keywords:** human MC2R, MRAP1, activation, trafficking, ACTH

## Abstract

Human melanocortin-2 receptor (hMC2R) co-expressed with the accessory protein mouse (m)MRAP1 in Chinese Hamster Ovary (CHO) cells has been used as a model system to investigate the activation and trafficking of hMC2R. A previous study had shown that the N-terminal domain of mMRAP1 makes contact with one of the extracellular domains of hMC2R to facilitate activation of hMC2R. A chimeric receptor paradigm was used in which the extracellular domains of hMC2R were replaced with the corresponding domains from *Xenopus tropicalis* MC1R, a receptor that does not interact with MRAP1, to reveal that EC2 (Extracellular domain 2) is the most likely contact site for hMC2R and mMRAP1 to facilitate activation of the receptor following an ACTH binding event. Prior to activation, mMRAP1 facilitates the trafficking of hMC2R from the ER to the plasma membrane. This process is dependent on the transmembrane domain (TM) of mMRAP1 making contact with one or more TMs of hMC2R. A single alanine substitution paradigm was used to identify residues in TM4 (i.e., I^163^, M^165^), EC2 (F^167^), and TM5 (F^178^) that play a role in the trafficking of hMC2R to the plasma membrane. These results provide further clarification of the activation mechanism for hMC2R.

## 1. Introduction

The activation of most melanocortin receptors (MCRS) such as MC1R, MC3R, MC4R, and MC5R is fairly uniform and can be achieved via the binding of melanocortin peptides such as ACTH, αMSH, βMSH, or γMSH to the respective receptor with varying degrees of efficacy. The exception is MC2R [1,2]. For example, human (h) MC2R can only be activated by ACTH, but not by any of the smaller MSH-sized ligands [2,3]. In addition, hMC2R has an obligatory requirement for interaction with the accessory protein, MRAP (melanocortin-2 receptor accessory protein), which will be referred to as MRAP1 to distinguish this accessory protein from its paralog MRAP2, to facilitate not only trafficking of the receptor from the endoplasmic reticulum to the plasma membrane, but also activation of the receptor at the plasma membrane following an ACTH binding event [4,5]. By comparison, the other hMCRs do not require MRAP1 to facilitate either activation or trafficking [1,6,7].

MRAP1 is a single pass transmembrane protein that forms an antiparallel homodimer with reverse topology [6,7]. In the case of humans, mutations in the *MRAP1* gene will result in Type II Familial Glucocorticoid Deficiency (FGD II) [7]. This accessory protein has three functional domains. The transmembrane domain (TM) facilitates the trafficking of hMC2R to the plasma membrane [5]. Within the N-terminal Domain, there is a reverse topology motif [5,8] which is required for the antiparallel orientation of the two MRAP1 monomers that comprise the mMRAP1 homodimer. Finally, in the N-terminal domain of mMRAP1 there is a four amino acid motif (L^18^ D^19^ Y^20^ L^21^) which serves as the activation motif [8]. Deletion of this motif, or alanine substitution of the motif will block activation of hMC2R expressed in CHO cells but does not block the trafficking of hMC2R to the plasma membrane [8]. It should be noted that MRAP2, a paralog of MRAP1, can facilitate the trafficking of hMC2R to the plasma membrane, but this accessory protein lacks the activation motif found in MRAP1 analogs, and as a result, cannot facilitate the activation of hMC2R [9]. The absence of an activation motif is a common feature of vertebrate MRAP2 orthologs [10].

A series of studies have used hMC2R and the accessory protein mMRAP1 as a model system to investigate the dynamics of the MC2R/MRAP1 interaction [5,8,11,12]. As noted, the L^18^ D^19^ Y^20^ L^21^ motif in the N-terminal of mMRAP1 [8] plays a key role in facilitating activation of hMC2R. However, since mMRAP1 forms a homodimer with reverse topology, the activation motif is present on the N-terminal domain of the homodimer facing the cytosol and the N-terminal domain of the homodimer facing the extracellular space. This orientation raised the question of whether MRAP1 interacts with an intracellular domain or an extracellular domain of hMC2R to facilitate activation. A recent study resolved this issue and showed that activation of hMC2R by mMRAP1 involves the extracellular N-terminal domain of the mMRAP1 homodimer making contact with an extracellular domain of hMC2R [12]. However, in the latter study, the extracellular domain involved in activation was not identified. The first objective of this study was to use a chimeric receptor paradigm in which the extracellular domains of hMC2R were individually replaced with the corresponding domain from *Xenopus tropicalis* MC1R to identify the extracellular domain on hMC2R that interacts with mMRAP1. The second objective used a single-alanine substitution paradigm of selected residues in TM4 and TM5 of hMC2R, in conjunction with a Cell Surface ELISA assay, to identify the transmembrane domain in hMC2R that interacts with the TM of mMRAP1 to facilitate the trafficking of the receptor to the plasma membrane.

## 2. Materials and Methods

### 2.1. Chimer Receptor Paradigm

To determine which extracellular domain of hMC2R interacts with mMRAP1, a set of chimeric receptors were made in which an extracellular domain of hMC2R was replaced with the corresponding domain from MC1R of the amphibian *Xenopus tropicalis* (xt). For example, the replacement of the N-terminal domain of hMC2R with the N-terminal domain of xtMC1R was designated hMC2R/NT xtMC1R. In a similar manner, individual chimer hMC2R receptors were made for EC1, EC2, and EC3 domains of hMC2R, and were designated hMC2R/EC1 xtMC1R, hMC2R/EC2 xtMC1R, and hMC2R/EC3 xtMC1R, respectively. The rationale for using xtMC1R sequences is presented in Results Section 3a (see Figure 1). The nucleotide sequences of the four hMC2R chimeric receptors are presented in Appendix A. A diagram of each chimeric hMC2R receptor is presented in Figure 2a.

### 2.2. DNA Constructs

Human MC2R (hMC2R; Accession #: AA067714.1), *Mus musculus* (mouse), Mrap1 (mMrap1; Accession #: NM_029844), *Xenopus tropicalis* Mc1r (xtMc1r; Accession#: XP 012817790), and all of the chimeric receptors (i.e., hMC2R/NT xtMC1R; hMC2R/EC1 xtMC1R; hMC2R/EC2 xtMC1R; hMC2R/EC3 xtMC1R) were synthesized by GenScript (Piscataway, NJ), and individually inserted into the pcDNA3.1+ expression vector. For cell surface ELISA assays, wild-type hMC2R and single alanine mutants of hMC2R in the TM4 domain (i.e., G^162^/A^162^; I^163^/A^163^; T^164^/A^164^; M^164^/A^164^; V^166^/A^166^; I^167^/A^167^), EC2 domain (F^168^/A^168^; H^170^/A^170^), and the TM5 domain (T^177^/A^177^; F^178^/A^178^; T^179^/A^179^; S^180^/A^180^; L^181^/A^181^; F^182^/A^182^; P^183^/A^183^) were synthesized with a N-terminal V-5 epitope tag, and individually inserted into a pcDNA3.1+ expression vector (GenScript). The cAMP reporter cDNA, CRE-Luc [13], was provided by Dr. Patricia Hinkle (University of Rochester, NY, USA).

### 2.3. ACTH and α-MSH Peptides

The melanocortin peptides used in this study were synthetic hACTH(1-24) and NDP-MSH purchased from Sigma-Aldrich Inc. (Saint Louis, MO, USA). These peptides were used in the cAMP CRE-Luciferase reporter gene assay at concentrations ranging from 10^−13^ M to 10^−6^ M.

### 2.4. Tissue Culture Procedure

Experiments were done utilizing Chinese Hamster Ovary (CHO) cells (ATCC, Manassas, VA, USA). The cells were grown in Kaighn’s Modification of Ham’s F12K media supplied by ATCC. Media was supplemented with 10% fetal bovine serum, 10 unit/mL penicillin, 100 µg/mL streptomycin, and 100 µg/mL normocin (Complete CHO media) The cells were grown in a 25 cm^3^ tissue culture flask with vent cap by CELLTREAT^TM^ (Pepperell, MA), and maintained in an incubator with 95% air, 5% CO_2_ at 37 °C. When the CHO cells reached 70–80% confluence, cells were split into new culture flasks using 0.05% trypsin/0.53 mM EDTA purchased from CORNING cellgro^TM^ (Corning, NY, USA). CHO cells were selected for this project due to the fact that this cell line does not express endogenous *mcr* genes [5,14], or endogenous *mrap* genes [15].

### 2.5. cAMP Reporter Gene Assay (CRE-Luciferase Assay)

For the cAMP Reporter gene Assay, 3.0 × 10^6^ cells/reaction were used. Cells were transfected with either *hMC2R* cDNA, chimeric *hMC2R/xtmc1r* constructs, or alanine-substituted *hMC2R* mutants (10 nm/transfection). All receptor constructs were co-transfected with *mMrap1* (30 nm/transfection), and the *cre-luciferase* construct (83 nmoles/transfection) [13]. Transfections were done utilizing the Amaxa Cell Line Nucleofector II system (Lonza Group, Basel, Switzerland) using program U-23. After a 10 min period of recovery the transfected cells were plated at a density of 1 × 10^5^ cells per well. After 48 h, the transfected cells were stimulated with hACTH(1-24) in serum-free CHO Media. Serial dilutions were carried out using hACTH(1-24) at concentrations ranging from 10^−7^ to 10^−13^ M. Each dose was tested in triplicate. The activation assay for xt*Mc1r* was done in an identical manner, and the transfected cells were stimulated with either hACTH(1-24) or NDP-MSH at concentrations ranging from 10^−6^ to 10^−12^ M.

Following a 4 h incubation at 37 °C, the stimulating solutions were removed and a luciferase substrate reagent (BrightGLO; Promega, Madison, WI, USA) was added to each well as described in Liang et al. [10]. A Bio-TEK Synergy HTX plate reader (Agilent Technologies, Santa Clara, CA, USA) measured the luminescence generated after a five-minute incubation period at room temperature. Transfected CHO cells incubated with serum-free media, but no ligand, were analyzed along with each experimental group to determine basal cAMP levels. Luminescence readings were corrected by subtracting the basal cAMP readings (serum-free media/no ligand) for each transfection dose response curve. The data for each dose response curve were fitted to the Michaelis–Menten equation to obtain EC_50_ values using Kaleidograph software (www.synergy.com). Data points are expressed as the mean ± SEM (*n* = 3). To analyze the level of activation, the data sets were analyzed using One-Way ANOVA followed by Tukey’s multi-comparison test using GraphPad Prism 2 software (GraphPad Inc, La Jolla, CA, USA). Significance was set at *p* ≤ 0.05.

### 2.6. Cell Surface ELISA

CHO cells were plated at 0.75 × 10^5^ cells/well in a fibronectin-treated 24-well culture dish and grown overnight in in a 37 °C CO_2_ incubator. Cells were transfected with cDNAs encoding *hMC2R-V5* alone (negative control)*, hMC2R-V5* + *mMrap1* (positive control), or *hMC2R-V5* alanine-substituted mutant constructs + *mMrap1* using jetPRIME transfection reagents (Polyplus transfection, Illkirch, France) as described in Barney et al. [16]. After 48-h, cells were fixed in 4% Paraformaldehyde, washed and then incubated with polyclonal V5-epitope antibody (1:500 dilution; Genetex, Irvine, CA, USA) followed by secondary HRP-conjugated goat anti-rabbit antibody (1:500 dilution). Cells were washed and treated with one-step 2,2′azinobis-3ethylbenzthiazoline-6-sulfonic acid (one-step ABTS) (Thermo Fisher Scientific, Waltham, MA, USA). Aliquots of supernatant were transferred to a 96-well plate and absorbance at 405 nm was measured using a Bio-TEK Synergy HTX plate reader (Agilent Technologies, Santa Clara, CA, USA). The rationale for selecting residues in TM4 and TM4 for single-alanine substitution is based on the alignment in Figure 3. The data presented in Figure 4 are normalized to the positive (hMC2R+MRAP1) and negative (hMC2R alone) controls, such that the positive control equals 100% and the negative control equals 0%. The normalized data were analyzed using a one-way ANOVA with Tukey’s multi-comparison post-test using GraphPad Prism software (GraphPad Inc, La Jolla, CA, USA), and the threshold for significance was set at *p* < 0.05 (Figure 4), or using Student’s *t*-test (Appendix A).

## 3. Results

### 3.1. Chimeric Receptor Analysis

To identify the extracellular domain of hMC2R that interacts with the extracellular N-terminal domain of the mMRAP1 homodimer, a chimeric receptor paradigm was used in which each extracellular domain of hMC2R was replaced with the corresponding extracellular domain of a melanocortin receptor that does not require interaction with MRAP1 for either activation or trafficking. The melanocortin-1 receptor (MC1R) of the amphibian, *Xenopus tropicalis* [17] fits these criteria. As shown in Appendix A, xtMC1R can be activated by either hACTH(1-24) or NDP-MSH, and co-expression of the receptor with mMRAP1 had no statistical effect, either positive or negative, on the sensitivity of the receptor to stimulation by either ligand (Appendix A). For stimulation with hACTH(1-24), the EC_50_ value for xtMC1R expressed alone was 6.8 × 10^−10^ M +/− 1.5 × 10^−10^, and the EC_50_ value when the receptor was co-expressed with mMRAP1 was 7.8 × 10^−10^ M +/− 9.3 × 10^−11^. These EC_50_ values are not statistically different (*p* = 0.84; One-Way ANOVA analysis). For stimulation with NDP-MSH, the EC_50_ value for xtMC1R expressed alone was 4.5 × 10^−11^ M +/− 1.4 × 10^−11^, and when the receptor was co-expressed with mMRAP1 the EC_50_ value was 8.6 × 10^−11^ M +/− 4.1 × 10^−11^. These EC_50_ values are also not statistically different (*p* = 0.95; One-Way ANOVA analysis). In addition, co-expression of xtMC1R with mMRAP1 had no statistical effect on the trafficking of the receptor to the plasma membrane as compared to CHO cells transfected with *xtMc1r* alone (*p* = 0.85; *n* = 3; Student *t*-Test; Appendix A).

An alignment of the amino acid sequences of hMC2R and xtMc1r is shown in Figure 1. The hypothetical membrane topology of hMC2R was predicted using the TMHMM program from the DTU Bioinformatics Server (https://www.bioinformatics.dtu.dk, accessed on 29 October 2018), and the two receptor sequences could be aligned by inserted two gaps into the hMC2Rsequence. While the primary sequence identity of the two receptors was only 40%, identical amino acid motifs were apparent in all seven transmembrane domains and the three intracellular domains, with the highest primary sequence identity observed for IC2 (81%). However, the primary sequence identity in the extracellular domains was more variable (N-terminal 21%, EC1 16%, EC2 0%, EC3 43%).

Based on the alignment presented in Figure 1, four chimeric receptors of hMC2R were made in which an extracellular domain from hMC2R was replaced with the corresponding domain from xtMc1r. The chimeric receptors were designated as: hMC2R/NT xtMC1R, hMC2R/EC1 xtMC1R, hMC2R/EC2 xtMc1r, and hMC2R/EC3 xtMc1r. The nucleotide sequences of *hMC2R*, *xtmc1r*, and the four chimeric receptors are presented in Appendix A.

It should be noted that hMC2R has C^21^ in the N-terminal domain and has C^254^ in EC3 that apparently form a disulfide bridge that is essential for the functionality of hMC2R [18]. At the corresponding sites in xtMC1R the cysteine residues are absent and instead D^24^ (N-terminal) and S^269^ (EC3) occupy these positions. The absence of the cysteine residues at these positions has no apparent effect on the functional expression of xtMC1R in CHO cells as seen in Appendix A. However, in our pilot experiments, we found that chimeric receptor hMC2R/NT D^26^ xtMC1R and chimeric receptor hMC2R/EC3 S^269^ xtMC1R, when co-expressed with mMRAP1 in CHO cells, could not be stimulated at any of dose of hACTH(1-24) (i.e.,10^−13^ M to 10^−7^ M; data not shown). As a result, the chimeric receptor hMC2R/NT xtMC1R was made with a cysteine residue at position 26, and the chimeric receptor hMC2R/EC3 xtMC1R was made with a cysteine residue at position 253 in the chimeric receptor (Appendix A). A diagram of the four chimeric receptors is presented in Figure 2A.

For the chimeric receptor analysis, hMC2R and the four chimeric receptors were individually co-expressed with mMRAP1 in CHO cells as described in Methods. As shown in Figure 4B, the dose response curves for the EC1 chimeric receptor, the EC2 chimeric receptor, and the EC3 chimeric receptor produced Vmax values similar in magnitude to the Vmax for hMC2R (Appendix A). However, the NT chimeric receptor (hMC2R/NT xtMC1R) responded in a robust manner to stimulation with hACTH(1-24) (Figure 2B) with a Vmax value 55% higher than the Vmax value for hMC2R (Appendix A). In addition, this chimeric receptor was as sensitive to stimulation by the ligand as hMC2R based on the statistical analysis (Figure 2C). From the perspective of ligand sensitivity, substitution at the N-terminal domain had no negative effect on the chimeric receptor relative to the response of hMC2R to stimulation.

The EC1 chimeric receptor (hMC2R/EC1 xtMC1R) and the EC3 chimeric receptor (hMC2R/EC3 xtMC1R) both had EC_50_ values that were approximately 10-fold lower than the EC_50_ value for hMC2R (Figure 2B). However, the decline in ligand sensitivity for both chimeric receptors was not statistically different from the EC_50_ value for hMC2R (Figure 2C).

The most dramatic effect was observed for the EC2 chimeric receptor (hMC2R/EC2 xtMC1R). The EC2 chimeric receptor was nearly three orders of magnitude less sensitive to stimulation by hACTH(1-24) as hMC2R (Figure 2C), and this shift in sensitivity was statistically significant from the EC_50_ value for hMC2R. These data would suggest that EC2 is the most likely contact site between hMC2R and the N-terminal domain of mMRAP1.

### 3.2. Single-Alanine Analysis of Residues in TM4, EC2, and TM5: Effects on Trafficking

Based on the outcome of the chimeric receptor analysis, either TM4 or TM5 of hMC2R are candidates as the potential contact site with the TM of mMRAP1 to facilitate the trafficking of the receptor from the ER to the plasma membrane. To address this issue, a single-alanine substitution paradigm was used, and the effect of the mutant hMC2R receptors on trafficking was evaluated using a Cell Surface ELISA protocol. The operating hypothesis was that hMC2R would have unique amino acid motifs in one or both of these domains that are not present in the other human MCRs. To this end, the TM4/EC2/TM5 domain for the human MCRs were aligned (Figure 3A). This analysis indicated that for the TM4 domain, 38% of the positions are identical (highlighted in black) or similar (gray). However, when the identity/similarity analysis was done for only hMC1R, hMC3R, hMC4R and hMC5R the sequence identity/similarity (highlighted in green) for these four receptors was 68% (Figure 3B). Figure 3B suggested that positions 162 to 167 might be reasonable targets to investigate TM4-mediated membrane trafficking, and single-alanine mutants were made for G^162^, I^163^, T^164^, M^165^, V^166^, and I^167^ (Figure 4A,B).

For TM5 the primary sequence identity/similarity for all five receptors was 29% (Figure 3A), however, when the analysis was done for positions in just hMC1R, hMC3R, hMC4R and hMC5R (highlighted in green) the sequence identity/similarity for these four receptors was 71% (Figure 3B), and positions 177 to 183 stood out as unique to hMC2R with the exception of F^182^ which is found in all human MCRs. To investigate TM5-mediated membrane trafficking, single-alanine mutants were made for T^177^, F^178^, T^179^, S^180^, L^181^, F^182^, and P^183^ (Figure 4C,D).

Finally, previous studies had shown that alanine replacement at F^178^ [19] or H^180^ [2] significantly decreased the activation of these mutant forms of hMC2R using a cAMP assay, or a cAMP reporter gene assay, respectively. Since the EC2 domain should not be involved in trafficking, these alanine mutants were viewed as positive controls for the Cell Surface ELISA assay (Figure 4E).

The results of the Cell Surface ELISA assays are presented in Figure 4 and the data set used to generate these graphs and the results of the One-Way ANOVA analyses appear in Appendix A. For the TM4 domain, three single-alanine mutants (I^163^/A^163^; M^165^/A^165^; V^166^/A^166^; Figure 4A,B) resulted in a significant decrease in trafficking relative to the positive control (Appendix A).

By contrast in the TM5 domain, only alanine substitution at F^178^ resulted in a statistically significant decrease in trafficking relative to the positive control (Figure 4C,D; Appendix A). In addition, alanine substitution at H^180^ in EC2 had no negative effect on trafficking. However, alanine substitution at F^168^ in EC2 resulted in a statistically significant decrease in trafficking relative to the positive control (Figure 4E; Appendix A).

A comparison of the primary sequences for the TM4/EC2/TM5 domains of MC2R orthologs from a broad spectrum of vertebrates (i.e., mammal, bird, bony fish) is presented in Figure 4F. The primary sequence identity/similarity for the TM4 and TM5 domains for these orthologs was 62% and 68%, respectively. In addition, the non-mammalian orthologs have F^168^ in their predicted EC2 domain, and F^178^ in their predicted TM5 domain. Note that these two residues are not found present in xtMC1R (Figure 1) and co-expression with mMRAP1 was not required for the trafficking of xtM1R to the plasma membrane (Appendix A).

## 4. Discussion

This study used a chimeric receptor paradigm to show that the EC2 domain of hMC2R is the most likely contact site with the N-terminal of mMRAP1 to facilitate activation (Figure 2). In addition, the Cell Surface ELISA analysis indicated that alanine substitutions at I^163^, M^165^, and V^166^ in TM4, F^168^ in EC2, and F^178^ in TM5 all decreased trafficking which would suggest that these residues are interacting with the TM domain of the mMRAP1 homodimer to facilitate trafficking of the receptor to the plasma membrane (Figure 4). In an earlier study, Chen et al. [19] observed that alanine substitution at F^168^ and F^178^ resulted in a decline in the binding of ACTH and the production of cAMP when the F^168^/A^168^ and F^178^/A^178^ mutant forms of hMC2R were expressed in OS3 cells. The results of the current study indicate that the apparent decline in binding and activation activity observed in the Chen et al., [19] study was most likely the result of a decline in the trafficking of the mutant hMC2R receptors to the plasma membrane. With respect to the location of F^168^ in hMC2R (i.e., TM4 or EC2), our analysis of the hypothetical membrane topology for hMC2R (Figure 1) positioned F^168^ as the first residue in EC2, but perhaps this residue is actually the last residue in TM4.

A diagram summarizing the results of this study and the predicted location of the binding site for the “message” motif (i.e., HFRW) of ACTH (25) is presented in Figure 5A,B.

Building off of the structure/function analysis done by Pogozheva et al., [21] on the HFRW binding site for hMC4R, which was confirmed by x-ray crystallographic analysis [22], Chen et al. [19] found that alanine replacement at E^80^ in TM2, D^103^ and D^107^ near or in TM3, and F^236^ and H^239^ in TM6 all decreased activation of hMC2R expressed in OS3 adrenal tumor cells. All of these residues appear to be part of the HFRW binding site for hMC2R and these residues are shown in the diagrams in Figure 5. In fact, these five residues are present in the MC2R orthologs from representatives of all the classes of vertebrates that have been analyzed [23,24]. However, it should be noted that ACTH also has an “address” motif (KKRR), that is needed for activation of MC2R orthologs [25], and this proposed binding site is predicted to involve EC2 and the N-terminal of MRAP1.

In this scenario H^170^ in the EC2 domain (Figure 5) plays an important role. Chung et al., [20] did an analysis of single position mutations in the *MC2R* gene and identified a patient with a mutation at H^170^ in EC2 that interfered with the patient’s ability to produce cortisol. Pharmacological studies indicate that substitution of an alanine residue at H^170^ results in an altered form of hMC2R that when co-expressed in CHO cells with mMRAP1 has a much lower sensitivity to stimulation by ACTH than wild-type hMC2R [20]. Since in the current study the hMC2R H^170^/A^170^ mutant did not interfere with trafficking (Figure 4E), perhaps the function of H^170^ is to stabilize the N-terminal domain of mMRAP1 so that EC2 and a portion of the N-terminal of mMRAP1 can serve as the binding site for the KKRR motif in ACTH. In an earlier study, Fridmanis et al. [26] did an extensive chimeric receptor analysis of hMC2R and postulated that the N-terminal of MRAP1 may be the KKRR binding site for ACTH. This scenario would explain why alanine substitution at the activation motif of mMRAP1 blocks the activation of the hMC2R/mMRAP1 heterodimer [8] but does not interfere with trafficking, and also why MRAP2 orthologs, which all lack an activation motif, cannot facilitate the activation of bony vertebrate Mc2r orthologs but can facilitate trafficking [19].

## 5. Conclusions

In summary, several studies point to the TM4/EC2/TM5 domain of hMC2R in conjunction with the N-terminal of MRAP1 as the possible location for the binding site for the “address motif” in ACTH [19,20,26]. The trafficking of hMC2R from the ER to the plasma membrane appears to involve the TM domain of mMRAP1 interacting with residues in TM4 and TM5 of hMC2R, and at least some of these residues involved in trafficking in the two TMs of hMC2R have been identified. However, an issue that is not resolved is an explanation for why α-MSH cannot activate hMC2R or apparently bind to the receptor [3,25,27]. The absence of the KKRR motif at the C-terminal terminal of α-MSH is given as an explanation for this observation. However, cartilaginous fish Mc2r orthologs have the same critical amino acids in TM2, TM3, and TM6 as bony vertebrate Mc2r orthologs, yet the cartilaginous fish Mc2r orthologs can be activated by either ACTH or α-MSH with varying degrees of efficacy [15,28,29]. Rather than primary sequence, perhaps the issue involves tertiary structure. The formation of the hMC2R/mMRAP1 heterodimer at the ER presumably prevents hMC2R from misfolding and then being degraded [5]. The diagram present in Figure 5C proposes the scenario that when the hMC2R/mMRAP1 heterodimer forms, the HFRW binding site (i.e., TM2, TM3, TM6) is in a closed position. In this scenario, at the plasma membrane the binding of the address motif of ACTH to the TM4-EC2-TM5/MRAP1 region of the heterodimer would result in a conformational change that opens the HFRW binding site to allow the binding of the message motif of ACTH and initiate activation of the receptor. To provide supporting evidence for the preceding scenario, structural modeling analyzes of the hMC2R/MRAP1 heterodimer is required, and that type of analysis is beyond the scope of the present investigation. To investigate possible conformational changes in the MC2R/MRAP1 heterodimer, such an effort would need to first produce a model of the MRAP1 homodimer that correctly captures the reverse topology of the accessory protein. In the meantime, binding studies taking advantage of the several mutant constructs of hMC2R and mMRAP1 that are currently available may be informative in understanding the complex activation of hMC2R. In this regard, a better understanding of the mechanism of how ACTH activates hMC2R may open the door to the design of pharmacological products that could have important clinical and scientific relevance.

## Figures and Tables

**Figure 1 biomolecules-12-01422-f001:**
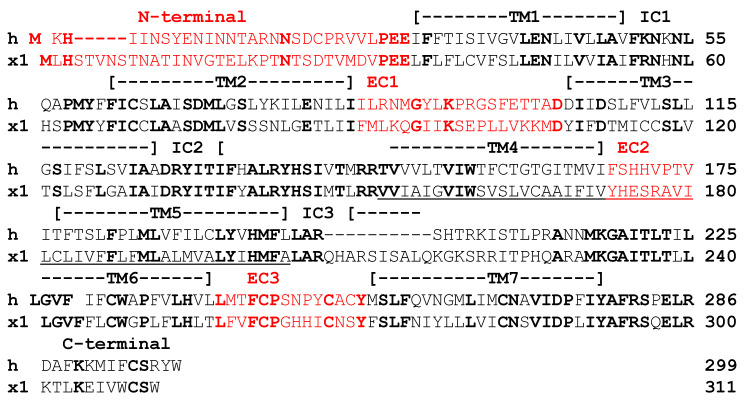
Alignment of Amino Acid Sequences of hMC2R and xtMC1R. The hypothetical membrane topology of hMC2R was predicted using the TMHMM program (https://www.bioinformatics.dtu.dk, accessed on 18 October 2018). The amino acid sequence of xtMC1R was aligned to hMC2R by inserting two gaps. The predicted extracellular domains of hMC2R and xtMC1R are highlighted in red. Identical positions in both receptors are in bold red for residues in an extracellular domain and bold black for residues in a TM, IC, or C-terminal domain. Abbreviations: h (human), x (*Xenopus tropicalis*), EC (extracellular domain, IC (intracellular domain), TM (transmembrane domain).

**Figure 2 biomolecules-12-01422-f002:**
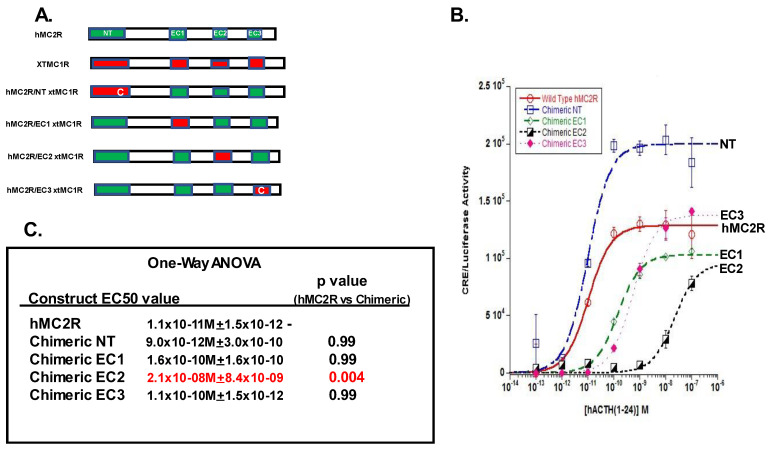
cAMP Reporter Gene Assay: Analysis of hMC2R/xtMC1R Chimeric Receptors. (**A**) The diagram shows the relative location of the extracellular domains present in hMC2R (**green**) and the corresponding domains present in xtMC1R (**red**), and the organization of the chimeric receptors. Note the insertion of a cysteine residue in hMC2R/NT xtMC1R and hMC2R/EC3 xtMC1R. Abbreviations: C (cysteine), NT (N-terminal domain), EC1 (extracellular domain 1), EC2 (extracellular domain 2), EC3 (extracellular domain 3). (**B**) Dose response curves for wild-type hMC2R, hMC2R/NT xtMC1R (Chimeric NT), hMC2R/EC1 xtMC1R (Chimeric EC1), hMC2R/EC2 xtMC1R (Chimeric EC2), and hMC2R/EC3 xtMC1R (Chimeric EC3) all co-expressed with mMRAP1 in CHO cells as described in Methods, and stimulated with hACTH(1-24). (**C**) The EC_50_ value for each dose response curve is presented (*n* = 3) and the results of One-way ANOVA analysis of the EC_50_ values. A *p* value < 0.05 is highlighted in red.

**Figure 3 biomolecules-12-01422-f003:**
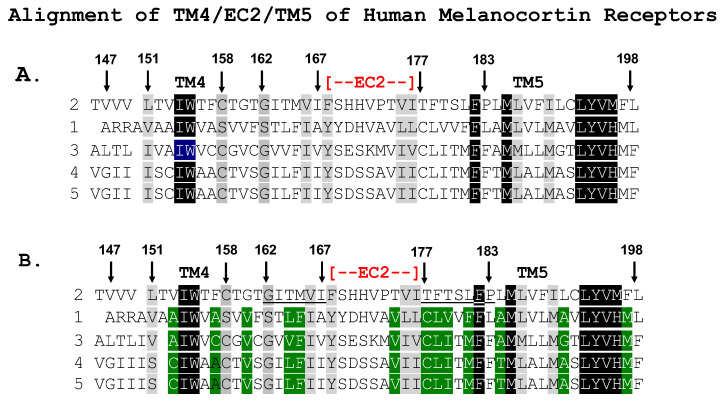
Alignment of Amino Acid Sequences of TM4/EC2/TM5 Domains of Human MCRs: Identity/Similarity Analysis. (**A**) The amino acid sequences of the TM4/EC2/TM5 domains for hMC2R (AA067714.1), hMC1R (Q01726.2), hMC3R (AKI72215.1), hMC4R (NP_005903.2), and hMC5R (NP_005904.1) were analyzed for primary sequence identity/biochemical similarity using BLOSUM (https://www.ncbi.nlm.nih.gov/Class/FieldGuide/BLOSUM62.txt, accessed on 16 August 2022). Positions that are heighted in black for all five receptors are identical. Positions that are similar for all five receptors are highlighted in grey. (**B**) The same analysis was done for only hMC1r, hMC3R, hMC4R, and hMC5R, and the positions that are identical or similar but not found in hMC2R are highlighted in green.

**Figure 4 biomolecules-12-01422-f004:**
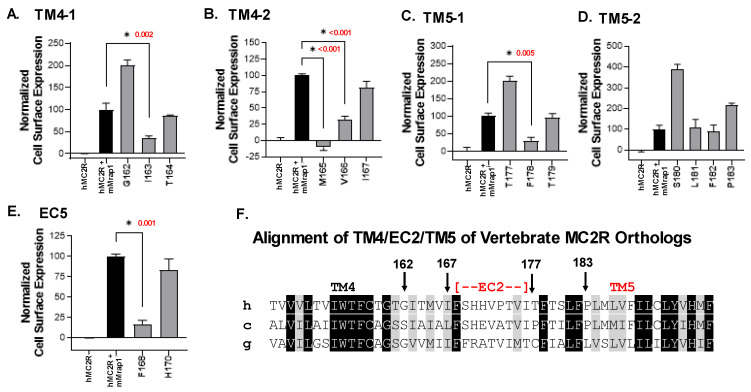
Cell Surface ELISA Analysis of Single-Alanine mutants of hMC2R. The Cell Surface ELISA analysis was performed as described in Methods. The negative control was hMC2R expressed alone. The positive control was hMC2R co-expressed with mMrap1. All of the single alanine mutant forms of hMC2R were co-expressed with mMRAP1. The one-way ANOVA *p* values for mutants that results in a decrease in trafficking relative to the positive control are shown in red. (**A**) Analysis of TM4 single alanine mutants G^162^/A^162^, I^163^/A^163^, and T^164^/A^164^. (**B**) Analysis of TM4 single alanine mutants M^165^/A^165^, V^166^/A^166^, and I^167^/A^167^. (**C**) Analysis of TM5 single alanine mutants T^177^/A^177^, F^178^/A^178^, and T^179^/A^179^. (**D**) Analysis of TM5 single alanine mutants S^180^/A^180^, L^181^/A^181^, F^182^/A^182^, and P^183^/A^183^. (**E**) Analysis of EC2 single alanine mutants F^168^/A^168^ and H^170^/A^170^. (**F**) Alignment of the TM4/EC2/TM5 domains of human MC2R (h), *Gallus gallus* (chicken; c) Mc2r, and *Lepisosteus osseous* (gar; g) Mc2r. As described in the legend to Figure 3, amino acid positions that are identical are highlighted in black, and amino acid positions that are highlighted in grey are similar based on BLOSUM analysis. * indicates statistical decrease in trafficking relative to the positive control.

**Figure 5 biomolecules-12-01422-f005:**
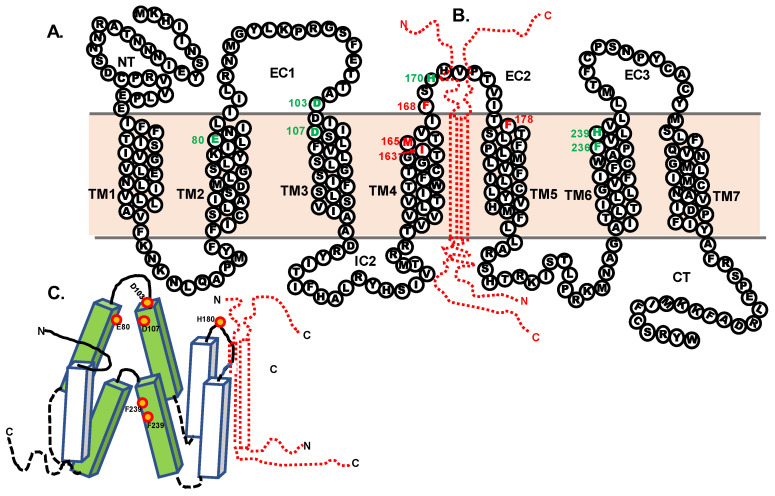
(**A**) This figure is a two-dimensional view of hMC2R based on the amino acid sequence presented in Figure 1. Amino acid positions highlighted in green are important for activation [4,6]. Amino acid positions highlighted in red play a role in trafficking (Figure 4). (**B**) A cartoon of mMRAP1 (red; not drawn to scale) is superimposed between TM4 and TM5. It is intended to show the orientation of mMRAP1 relative to hMC2R in the hMC2R/mMRAP1 heterodimer. (**C**) This cartoon depicts the predicted conformation of hMC2R with a “closed” HFRW binding site prior to an ACTH binding event. A diagram of mMRAP1 (**red**) is superimposed at the TM4/EC2/TM5 domain. The relative position of critical amino acid positions predicted to be involved in activation are highlighted in orange [19,20].

## Data Availability

All data pertaining to this study are stored in the laboratory of Robert M. Dores (robert.dores@du.edu) and are available upon request.

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
