# Peer review of "Human Melanocortin-2 Receptor: Identifying a Role for Residues in the TM4, EC2, and TM5 Domains in Activation and Trafficking as a Result of Co-Expression with the Accessory Protein, Mrap1 in Chinese Hamster Ovary Cells"

_biomolecules, 2022, doi:10.3390/biom12101422_

Round 1

Reviewer 1 Report

The authors presented new information the important role of TM4/EC2/TM5 domain of hMC2R in activation and trafficking, resulting in the increasing interest and data on activation mechanism for MC2R.

I have the following recommendations:

1. What I'm most concerned about is why did the author select the combination using three different species genes, human MC2R, mouse Mrap1, and Xenopus tropicalis mc1r?

Human MRAP1 also does not alter the hMC1R trafficking. The present study showed that mouse MRAP1 has no effect on Xenopus tropicalis MC1R trafficking. An alignment of the amino acid sequences also showed significant differences between the hMC2R and xtMC1R. This may also be the reason for the use of a different species. Using human MC1R and human MRAP1 might be a better manner. In addition, in Figure 3, Alignment of Amino Acid Sequences of all human MCRs was analyzed, not included xtMC1R. I am confused.

2. Gene names cannot be abbreviated with the species name. For human, gene names should be in italic uppercase (like MC4R); In mouse, the first letter is italic uppercase, and the rest is italic lowercase (like Mc4r); For other species, gene names are in italic lowercase (like mc4r). 

Protein names should be in uppercase, such as Figure 2 (Mc3r, Mc4r, and Mrap2 need change to MC3R, MC4R, and MRAP2, respectively).

3. The Result need to be simplified and mainly focused the current results. All of paragraphs 1 to 6 (total 8) described the background, and only two talked the present results.

4. Figure 1 is in garbled state and may need to be redone.

5. Figure 4, from the pane A to E, do all of these panes use the same negative and/or positive controls? Why are the values so different? Did the signaling pathways in these Single-Alanine mutants of hMC2R do?

6. Other article (PMID: 25074265) also reported the importance role of the second extracellular loop of MC2R in MC2R activation

Reviewer 2 Report

The authors present a very interesting study on the hMC2R extracellular (EC) and transmembrane (TM) domains involved in interaction with mMRAP1 that are important for the activation and the trafficking of the hMC2R. Using chimera between hMC1R domains and MC2R they found that the EC2 may be the contact site for hMRAP1 involved in ACTH activation of hMC2R.

They further studied some amino acids in the same EC2 domain as well as surrounding domains in TM4 and TM5 to see whether they affect activation or trafficking. They found that the A170 substitution seems to play a role in activation and not in trafficking.

The statistical test should be used cautiously when so few samples (3 technical replicates each) are used. 

The writing of the abstract, introduction would be improved with better introducing the concept of the chimeric receptor paradigm. The discussion could be shortened with paragraphs that concern only this study. The figures should have a concise description and be simplified if possible and graphical quality can be improved.

Minor comments

Abstract

L16: what do the authors mean by “chimeric receptor paradigm”? this could be explained earlier instead at the beginning of the results (L145) introduction or abstract.

Introduction

L44-45: the sentence is not clear.

Materials and Methods

L74: Could the authors better explain the construct they design: for example NT = N-terminal? hMC2R/NT xtMC1R is it the fusion between the N-terminal part of MC2R with full length MC1R and the other one are the proteins full length? Or mention the table S1.

L75: hMC2R/EC3 xtMc1 is a R missing

L105: Basel not *Basil*.

L122: To what corresponds “the corrected data sets”?

L133: It may be interested to give the used antibodies dilutions for the assay if somebody wants to repeat it.

Figure S1: 

Caption: The results should be in the results section.

L227: give the statistical value.

Figure 2: in the part B, the EC50 could be removed and place in the C figure instead of the statistical values.

Figure 4: How is explained the variation in expression levels of the positive controls between experiments? Are the results reproductible between experiments? Maybe the results should be presented as percentage of the positive control after extracting the negative result of each test. Were the absorbance results control with immunofluorescence?

L286-287: in the figure, neg and pos can be replaced by hMC2R, hMC2R + mMRAP1 for better visibility

Discussion

the first three paragraphs do not bring information for the discussion. They should be removed or added to the introduction. 

Round 2

Reviewer 1 Report

The authors have made sufficient changes according to the suggestions.